# Diabetes, Iron-Deficiency Anemia, and Endocrine, Nutritional, and Metabolic Disorders in Children: A Socio-Epidemiological Study in Urban Kazakhstan

**DOI:** 10.3390/ijerph22091346

**Published:** 2025-08-28

**Authors:** Svetlana Rogova, Olga Plotnikova, Marat Kalishev, Karina Nukeshtayeva, Zhanerke Bolatova, Aza Galayeva

**Affiliations:** 1School of Public Health, Karaganda Medical University, Karaganda 100008, Kazakhstan; s.rogova@qmu.kz (S.R.); kalishev@qmu.kz (M.K.); galaeva.a@qmu.kz (A.G.); 2Department of Occupational Hygiene and Occupational Pathology, Omsk State Medical University, Omsk 644099, Russia; olga.plotnikova7@mail.ru

**Keywords:** anemia, diabetes, children, Kazakhstan, nutrition

## Abstract

This study analyzes ten-year trends in the incidence of iron-deficiency anemia (IDA), diabetes mellitus (DM), and endocrine, nutritional, and metabolic disorders (ENMDs) among children and adolescents (0–17 years) in urban areas of Kazakhstan, considering socio-economic influences. A retrospective analysis of national data from 2013 to 2023 was conducted using linear regression to assess temporal trends and associations with health and economic indicators. Nationally, IDA incidence declined significantly: –278.4 cases per 100,000 among children aged 0–14 and –305.4 among adolescents aged 15–17 (both *p* < 0.001). ENMD incidence also decreased, particularly among adolescents (–154.0 per 100,000; *p* < 0.001). A 1000 KZT increase in household food expenditures was associated with a reduction in IDA incidence by 203–216 cases per 100,000 (*p* < 0.001), emphasizing the importance of accessible, nutritious diets. In contrast, DM incidence among adolescents rose by 1.7 cases annually per 100,000 (*p* < 0.05), possibly reflecting urbanization, lifestyle changes, and increasing obesity. DM and ENMD rates were significantly linked to consumption expenditures, pediatric bed availability, and endocrinologist density. These findings underscore the need for integrated, equity-focused prevention and improved healthcare access for children and adolescents amid ongoing demographic and nutritional transitions.

## 1. Introduction

A fundamental principle of state social policy in most countries around the world is the preservation and promotion of population health [1,2,3]. The health of the younger generation is a crucial factor in ensuring quality of life, sustainable development, and the realization of a country’s socio-economic potential [4,5]. In the modern context, child health is viewed as a complex, multidimensional indicator shaped by both internal processes—such as urbanization, industrialization, and demographic shifts—and external socio-economic factors, including globalization, migration, and changes in the global epidemiological landscape [6,7].

Findings from international studies, including those conducted by WHO and UNICEF [5,8,9,10], confirm that childhood and adolescence are critical periods during which the foundations of lifelong health are established. Moreover, the risk of developing chronic noncommunicable diseases in adulthood is largely associated with lifestyle and environmental conditions during early life [11].

Health indicators among children and adolescents are among the most objective and sensitive criteria used to assess public health at the population level [12,13]. Their comprehensive analysis, employing modern epidemiological and statistical methods, makes it possible to identify current trends, regional disparities in population health, predict potential epidemiological risks, and evaluate the effectiveness of ongoing public health programs [13,14,15,16,17].

Contemporary approaches in public health emphasize the key roles of nutrition, access to medical care, and the level of urbanization in shaping the health of children and adolescents [18,19]. Urbanization, as one of the leading socio-economic transformation processes in developing countries, exerts a significant influence on lifestyles, dietary patterns, and behavioral models among the younger population [20,21]. For example, Popkin and Reardon [22] have shown that urbanization contributes to increased consumption of high-calorie, low-nutrient foods, which is directly linked to rising rates of obesity and metabolic disorders.

Despite recent positive shifts in Kazakhstan’s macroeconomic and social spheres—including a steady decline in poverty rates, changes in household spending patterns, and improved access to healthcare services—child health indicators still exhibit marked regional disparities [23,24]. These differences are likely driven by a combination of social determinants, such as the level of urbanization, dietary practices, and inequality in access to medical care.

A review of the scientific literature reveals that Kazakhstan still lacks published, comprehensive, long-term studies reflecting trends in child and adolescent morbidity under conditions of accelerated urbanization and lifestyle transformation. Existing publications are generally fragmented and limited in both time span and geographic coverage.

Previously, Tretyakova et al., based on data from 1999 to 2018, reported a substantial increase in primary morbidity among children (by 34.7%) and adolescents (by 40.1%) [25]. However, that study did not include data from the most recent five years and did not analyze trends by disease categories related to nutrition and metabolic disorders, leaving a significant gap for further investigation.

This is especially relevant for conditions classified under ICD-10 Chapter IV, “Endocrine, nutritional, and metabolic diseases,” as well as iron-deficiency anemia and diabetes mellitus, which in this context are considered key indicators of nutritional status and overall child health [26]. The selection of these disease categories is justified by their sensitivity to social determinants of health and the observed upward trends in their prevalence among children in rapidly urbanizing countries [18,21,22,27].

The ten-year analysis period (2013–2023) was chosen based on the availability of comparable statistical data, which became systematically collected following the implementation of a unified ICD-10-based reporting format within Kazakhstan’s national statistical system. Furthermore, the last decade has seen significant socio-economic changes in Kazakhstan, including healthcare reforms, accelerated urbanization, shifts in population dietary behavior, and improved access to medical services [28,29,30]. These factors underscore the relevance of examining this specific period to identify long-term trends in child morbidity.

Therefore, the aim of this study is to conduct a comprehensive assessment of long-term (2013–2023) trends in morbidity among children and adolescents (aged 0–17 years) for selected disease groups, including endocrine, nutritional, and metabolic disorders, as well as iron-deficiency anemia and diabetes mellitus, with consideration of key socio-economic determinants of health. This analysis covers both national and regional data for urban populations in two age cohorts, helping to fill an existing knowledge gap, identify current epidemiological trends, and provide an evidence base for designing effective preventive measures and improving child health programs in the context of ongoing socio-economic change.

## 2. Materials and Methods

We conducted a retrospective data analysis using secondary data covering the period from 2013 to 2023. The dataset included 15 regions and 2 cities (Astana and Almaty). Quantitative data on socio-economic and healthcare indicators were extracted. Variables concerning population health and health indicators were extracted from the statistical yearbooks “Health of the population of the Republic of Kazakhstan and activities of health care organizations” for 2013–2023 and include the following [31]:

Incidence of endocrine, nutritional, and metabolic diseases (ENMDs) includes thyroid disorders, type 1 diabetes mellitus, type 2 diabetes mellitus, malnutrition, obesity, and metabolic syndromes;

Incidence of diabetes mellitus (DM), which focuses on chronic hyperglycemia, is subdivided into insulin-dependent diabetes, non-insulin-dependent diabetes, malnutrition-related diabetes, and other specified forms;

Incidence of iron-deficiency anemia (IDA), which specifically refers to anemia caused by inadequate iron intake or absorption, including nutritional iron deficiency;

Number of specialized therapists per 10,000 urban population (number of specialized therapists);

Number of endocrinologists per 10,000 urban population (number of endocrinologists);

Number of pediatricians per 10,000 urban population (number of pediatricians);

Number of organizations with children’s outpatient departments and offices per 1000 population;

Number of pediatricians (including neonatologists) per 1000 children aged 0–14 years;

Provision of children’s beds of all profiles per 1000 children aged 0–14 years (hospital beds).

The socio-economic indicators of the population of the Republic of Kazakhstan included in the analysis were extracted from regional and national statistical yearbooks and the Bureau of National Statistics of the Republic of Kazakhstan [32]. These indicators include the following:

The proportion of the urban population with incomes below the subsistence minimum (percentage);

The Gini coefficient, which quantifies income inequality on a scale from 0 (perfect equality) to 1 (maximum inequality);

Average monetary expenditures of households per capita (in thousand KZT);

Per capita consumption expenditures by region (in thousand KZT);

Average per capita monetary expenditures on food (in thousand KZT).

In the Republic of Kazakhstan, data collection on the primary incidence of type 1 and type 2 diabetes mellitus, iron-deficiency anemia, and other metabolic disorders among children is regulated by official directives of the Ministry of Health, particularly Order No. 128 dated 6 March 2013 (as amended), which approves the standardized forms of administrative health data [33]. Healthcare institutions are required to record newly diagnosed cases in accordance with ICD-10 classifications (including E10–E14 for diabetes and D50 for iron-deficiency anemia), specifying the patient’s age, region, and, where available, sex. Data collection is conducted by attending physicians during outpatient or inpatient visits and is entered into electronic medical information systems. Aggregated data are subsequently submitted to regional public health authorities and then to the Ministry of Health. Each year, by February 10, all healthcare providers must submit their summary reports using Form No. 128 as prescribed.

The calculation of primary morbidity indicators is carried out based on the methodology approved by the Ministry of Health, whereby the incidence is expressed as the number of newly registered cases per 100,000 children in the population [34]. Finalized data are published through official platforms such as the Bureau of National Statistics [32] and its specialized child health portal [35], as well as in the annual statistical compendium “Healthcare of Kazakhstan and Activities of Healthcare Organizations.” However, disaggregation of data by sex and age is sometimes limited, as some publicly available datasets are presented in aggregated form.

###  Data Analysis

A descriptive statistical analysis was conducted for all variables, including the calculation of means, standard deviations, minimum and maximum values. The normality of each variable was assessed using the Shapiro–Wilk test. Variables that did not meet the assumption of normal distribution were transformed using the natural logarithm. Indicators such as the proportion of the urban population with incomes below the subsistence minimum (%), the number of specialized general practitioners, and the number of endocrinologists were transformed using the natural logarithm. We applied linear regression analysis to examine temporal trends in childhood disease incidence across all regions of Kazakhstan from 2013 to 2023 and to quantify the average annual changes in the variables [36].

Subsequently, multiple linear regression analysis was performed as an ecological model, incorporating socio-economic and healthcare-related variables as predictors at the regional rather than individual level. The dependent variables included the incidence rates of IDA, DM, and ENMD among children aged 0–14 years and adolescents aged 15–17 years residing in urban areas. The following variables were included as independent predictors in the multiple linear regression models: the proportion of the urban population with incomes below the subsistence minimum (%), the Gini coefficient which quantifies income inequality on a scale from 0 (perfect equality) to 1 (maximum inequality), average household monetary expenditures per capita (thousand KZT), household income used for consumption by region (thousand KZT), and average per capita monetary expenditures on food (thousand KZT). Additionally, healthcare-related indicators were included: the number of specialized general practitioners, the number of general practitioners, the number of endocrinologists, and the number of pediatricians. Structural healthcare capacity was also considered through the number of healthcare organizations with pediatric outpatient departments per 1000 population, the number of pediatricians (including neonatologists) per 1000 children aged 0–14 years, and the availability of pediatric hospital beds of all profiles per 1000 children aged 0–14 years. Each model was constructed with a single dependent variable (the incidence of IDA, DM, or ENMD in either age group) and a set of predefined independent variables related to socio-economic status and healthcare availability. Model selection was based on theoretical relevance and empirical correlation, and all predictor variables were included simultaneously in the regression analysis. Multicollinearity was assessed using tolerance and variance inflation factor (VIF) statistics, ensuring that all included variables met acceptable thresholds (tolerance > 0.1; VIF < 4). Separate models were developed for each outcome and age group, resulting in six models in total.

## 3. Results

### 3.1. Trends in DM, IDA, ENMD Incidence of Children in Urban Areas of Kazakhstan

Table 1 presents the results of an analysis of the average annual changes in the primary incidence of IDA, DM, and endocrine, nutritional, and metabolic diseases among children aged 0–14 years across various urban regions of Kazakhstan.

Overall, there has been a significant decrease in the incidence of IDA in Kazakhstan, with an average annual decline of 285.52 cases per 100,000 population (*p* < 0.001). At the same time, there has been a moderate but statistically significant increase in the incidence of DM (0.84 per 100,000 population, *p* < 0.05) (Figure 1), while the incidence of ENMD has also significantly decreased (–75.68 per population, *p* < 0.001).

At the regional level, the patterns are heterogeneous. In the Akmola region, a significant decrease is observed in both IDA (–122.47 per 100,000 population, *p* < 0.05) and ENMD (–21.41 per 100,000 population, *p* < 0.01), while the change in DM incidence is not statistically significant. In the Aktobe region, a statistically significant decline is recorded across all three indicators, especially pronounced for IDA and ENMD. Some regions, such as the Almaty and Atyrau regions, demonstrate less pronounced and not always statistically significant changes. In the Almaty region, the trends across all three categories fail to reach statistical significance. In contrast, in the Atyrau region, the prevalence of DM increases significantly (0.79 per 100,000 population, *p* < 0.01), despite some decrease in IDA and ENMD.

The Zhambyl region stands out, showing a significant reduction in IDA (–357.01 per 100,000 population, *p* < 0.001) alongside a simultaneous significant increase in DM (2.21 per 100,000 population, *p* < 0.01). A similar situation is observed in the Pavlodar region, where DM incidence also rises (2.25 per 100,000, *p* < 0.01) against the background of a decrease in other indicators. The South Kazakhstan region demonstrates the most pronounced decrease in IDA (–591.3 per 100,000, *p* < 0.05) and a significant reduction in ENMD (–165.24 per 100,000 population, *p* < 0.001), while no significant changes are observed in DM incidence. In Almaty city, despite a significant decrease in ENMD (–274.13 per 100,000 population, *p* < 0.001), the trends in IDA and DM are weak and statistically non-significant.

The data presented in Table 2 reflect the average annual changes in the primary incidence of IDA, DM, and ENMD among adolescents aged 15–17 years in various urban regions of Kazakhstan. In contrast to the younger age group, this cohort exhibits both more pronounced negative trends for several conditions and localized increases, particularly in the incidence of DM.

At the national level, a stable and significant decrease in the incidence of IDA continues to be observed, with an average annual decline of 305.39 cases per 100,000 population (*p* < 0.001). The incidence of ENMD is also declining substantially (–153.92, *p* < 0.001). In contrast, DM shows a statistically significant increase, with an average annual rise of 1.67 cases (*p* < 0.05) (Figure 1).

At the regional level, both expected and unexpected deviations from the overall national trend are observed. In several regions—such as Aktobe, Kostanay, Zhambyl, and East Kazakhstan—the reduction in IDA reaches high statistical significance, and in most of these areas, it is accompanied by a decline in ENMD. Particularly sharp decreases were recorded in the Mangistau (–801.52, *p* < 0.001) and Aktobe (–748.51, *p* < 0.01) regions. At the same time, some regions exhibit mixed trends in DM. In the Kyzylorda region, a significant increase in DM was recorded (5.72, *p* < 0.05), and similarly in the North Kazakhstan region, where the annual increase reached 9.64 cases (*p* = 0.012)—the highest value among all regions. In Almaty city, a significant decrease is observed in both IDA (–381.56, *p* < 0.001) and ENMD (–297.35, *p* < 0.01), while DM rates remain stable, with no statistically significant changes. A similar pattern is observed in the Pavlodar region, where ENMD shows a particularly pronounced decline (–420.2, *p* < 0.001), although the increase in DM is not statistically confirmed. Some regions, such as the West Kazakhstan and Karaganda regions, demonstrate an increase in ENMD. In the case of Karaganda, the increase is statistically significant (55.76, *p* < 0.01), despite an overall favorable trend in IDA incidence.

### 3.2. Descriptive Statistics of Variables

In terms of ENMD incidences among children 0–14 years of age, the average rate was 1047.80 per 100,000, with a standard deviation of 738.54. Among 15–17-year-olds, the rate was almost twice as high at 2196.89, with a significant range of values (163.30 to 8234.90) and a high standard deviation (1447.70) (Table 3).

The mean of IDA among children 0–14 years old in urban areas was 2396.87 cases per 100,000, with a range of 253.00 to 7421.10 and a standard deviation of 1704.35. In adolescents 15–17 years of age, the similar rate was higher with an average of 2526.10 per 100,000, with a range of 111.40 to 9957.60 and a standard deviation of 1972.52.

The incidence of DM among children 0–14 years averaged 16.55 per 100,000, ranging from 2.50 to 60.90 cases (SD = 9.46). Adolescents 15–17 years had a higher rate of 23.65, with a wider range (0–166.70) and a standard deviation of 22.55.

The average share of the urban population with incomes below the subsistence minimum (poverty level) was 2.46%, with a standard deviation of 1.25%. The mean value of the Gini coefficient was 0.24, with a standard deviation of 0.04. The average household monetary expenditures per capita amounted to 55.20 thousand KZT, with a standard deviation of 20.70 thousand KZT. The average per capita income used for consumption was 57.16 thousand KZT, ranging from 24.13 to 121.4 thousand KZT, with a standard deviation of 20.62 thousand KZT. The average per capita food expenditures were 13.18 thousand KZT, with a standard deviation of 6.41 thousand KZT (Table 4).

The average number of general practitioners was 5.80 per 10,000, ranging from 1.80 to 10.90, with a standard deviation of 1.64. The average number of pediatricians in urban areas was 3.84 per 10,000, with a range from 2.10 to 8.10 and a standard deviation of 1.06. Endocrinologists were the least represented, with an average of 0.60 per 10,000, ranging from 0.20 to 1.40.

The average number of healthcare organizations with pediatric outpatient departments was 107.30 per 1000 population, with a notably wide range from 7.00 to 1337.00 and a standard deviation of 220.09.

The availability of pediatricians (including neonatologists) for the 0–14 age group averaged 0.823 per 1000 children, with a minimum value of 0.19 and a maximum of 2.65 (SD = 0.413). The average availability of pediatric hospital beds of all profiles for children aged 0–14 years was 3.99 beds per 1000 children, with a standard deviation of 1.16 and a range from 1.91 to 8.01.

### 3.3. Factors Affecting to the Incidences of IDA, DM, and ENMD Among Children Aged 0–14 and Adolescents Aged 15–17

Multiple linear regression analysis revealed a statistically significant negative association between average per capita food expenditures and the incidence of IDA (IDA) among children aged 0–14 years living in urban areas. According to the results, an increase in food expenditures by 1000 KZT was associated with a decrease in IDA incidence by an average of 203 cases per 100,000 children (Table 5).

A similar pattern was observed among adolescents aged 15–17 years: a statistically significant negative relationship was found between food expenditures and IDA incidence. An increase in food expenditures by 1000 KZT was associated with a decrease in IDA incidence by 216 cases per 100,000 adolescents.

Additionally, regression analysis demonstrated a statistically significant positive association between the number of pediatricians and the incidence of endocrine, nutritional, and metabolic diseases (ENMDs) among children aged 0–14 years. The regression coefficient was 890.99 with a standard error of 70.12, indicating an increase in ENMD incidence by 891 cases per 100,000 children for each additional pediatrician per 10,000 population.

Among adolescents aged 15–17 years, the number of specialized general practitioners and general practitioners also showed a statistically significant relationship with the incidence of ENMD. An increase of one specialized practitioner was associated with an increase in incidence by 604 cases per 100,000 adolescents. For general practitioners, the corresponding figure was 801.57 (SE = 319.88; *p* = 0.037).

Analysis of factors influencing the incidence of DM among children aged 0–14 years revealed significant associations with two predictors: average household monetary expenditures per capita and the number of general practitioners. Household expenditures were significantly and positively associated with DM incidence (B = 0.489; SE = 0.092; *p* = 0.001), indicating an increase of 0.489 cases per 100,000 children for every additional 1000 KZT spent. The number of general practitioners also had a significant positive effect on DM incidence (B = 5.259; SE = 1.504; *p* = 0.008), corresponding to an increase of 5.26 cases per 100,000 children for each additional practitioner per 10,000 population.

In the analysis of factors associated with DM incidence among adolescents aged 15–17 years, it was found that household income used for consumption, the availability of pediatric hospital beds, and the number of endocrinologists in urban population (expressed in natural logarithm) were significant predictors. Consumption-related income showed a positive effect on incidence (B = 0.379; SE = 0.038; *p* < 0.001), with an increase of 0.379 cases per 100,000 adolescents for every 1000 KZT increase in income. Additionally, an increase of one pediatric bed was associated with an increase in DM incidence by 19.3 cases (B = 19.314; SE = 2.677; *p* < 0.001). The log-transformed number of endocrinologists was also a significant factor (B = 56.508; SE = 14.876; *p* = 0.007): a 1% increase in the number of endocrinologists was associated with an average increase of 0.565 cases of DM per 100,000 adolescents.

## 4. Discussion

The conducted analysis provided a comprehensive assessment of the morbidity dynamics among children and adolescents (aged 0–17 years) in Kazakhstan over the past decade, focusing on endocrine, nutritional, and metabolic diseases (ENMDs), iron-deficiency anemia (IDA), and diabetes mellitus (DM), while accounting for key socio-economic determinants of health. The findings revealed diverse trends in disease prevalence across different regions and age groups, which carry important practical implications for improving preventive measures and strategically developing the pediatric healthcare system in the Republic of Kazakhstan.

Notably, a statistically significant decline in the primary incidence of IDA was observed at both national and regional levels among children and adolescents in both age categories (0–14 and 15–17 years), aligning with global trends. According to a global review [37], between 1990 and 2019, the age-standardized prevalence of anemia decreased by 13.4% across 204 countries and territories. A similar positive shift has been documented in Asian countries, where childhood anemia prevalence declined by 11.9% over the past three decades (1990–2021) [38]. The most pronounced improvements in IDA were observed in countries with high and medium socio-demographic development, due to a combination of factors: improved nutrition, implementation of iron deficiency prevention programs, increased access to healthcare, and enhanced monitoring of nutritional status in children and adolescents [38,39,40,41,42].

The results of this study indicate that Kazakhstan generally follows these global trends. According to the ecological regression analysis, over the period 2013–2023, an increase of 1000 KZT in per capita food expenditures was associated with an annual decrease of 203–216 cases of IDA per 100,000 children. This highlights the critical role of adequate and diverse nutrition as a major mechanism in anemia prevention.

However, the analysis also revealed substantial regional variation in the dynamics of IDA, which may be attributed to factors such as unequal economic development, differences in household income levels and coverage by prevention programs, as well as regional dietary habits. Migration patterns and intra-urban disparities in living conditions may also play a role, consistent with global findings linking anemia trends to social and economic determinants of health.

In contrast to the positive dynamics observed for IDA, the analysis of DM among children and adolescents in urban areas of Kazakhstan revealed more concerning trends. A statistically significant increase in primary DM incidence was recorded at the national level for both children aged 0–14 and adolescents aged 15–17, with the latter group showing a more pronounced annual growth rate.

These findings are consistent with global trends, which have shown a steady rise in pediatric DM incidence in recent decades [43,44,45,46,47]. For example, according to Hong YH et al., the prevalence of type 2 DM among youth in South Korea increased more than fourfold over 15 years [46]. In Germany, as reported by Stahl-Pehe et al., between 2002 and 2020, the standardized prevalence of type 1 DM in children rose by 77%, while type 2 DM nearly tripled (from 3.4 to 10.8 per 100,000), with the highest increase observed in adolescents aged 15–19 [48]. These increases are commonly attributed to urbanization, lifestyle changes, greater consumption of calorie-dense and processed foods, reduced physical activity, and the growing prevalence of overweight and obesity among children and adolescents [44,45,46,47,48,49].

A substantial body of evidence supports a physiological link between iron status and endocrine/metabolic function. For instance, iron deficiency impairs thyroid hormone synthesis by reducing thyroid-peroxidase activity and diminishing conversion of T_4_ to T_3_, which can lead to subclinical or overt hypothyroidism [50,51]. Similarly, Mendelian randomization studies indicate a causal association between hypothyroidism and increased risk of IDA (OR ~ 1.10, *p* < 0.001) [52]. Moreover, in pediatric populations with obesity and insulin resistance, disrupted iron homeostasis and chronic inflammation are tied to both IDA and metabolic dysregulation, suggesting a bidirectional relationship between iron deficiency and metabolic health outcomes [53]. These findings mirror our national observation that reductions in IDA were paralleled by declines in broader endocrine, nutritional, and metabolic disorders. The epidemiological and clinical evidence supports the hypothesis that improvements in iron status may contribute to systemic metabolic stabilization, and vice versa. We have added a brief discussion of these mechanistic and clinical insights in the revised manuscript to contextualize the link between IDA and ENMD trends.

The findings of this study underscore the significant role of socio-economic factors in shaping these trends. Regression analysis showed that DM incidence among children aged 0–14 was significantly associated with higher household monetary expenditures per capita. For adolescents aged 15–17, key influencing factors included consumption-related income, availability of pediatric hospital beds, and the number of endocrinologists. These results align with global data indicating that increased income and access to high-calorie, sugar- and fat-rich foods—particularly in urbanized settings with lifestyle shifts—contribute to higher exposure to dietary risk factors for DM among youth [39,42,44]. The positive correlation between medical resource availability (e.g., number of pediatric beds and endocrinologists) and reported DM incidence in adolescents likely reflects increased diagnostic activity in regions with better access to specialized care. Similar associations have been reported in international studies [43,46,49], where disparities in access to healthcare services contribute to regional variability in morbidity statistics. Thus, regional differences within Kazakhstan may be explained by both actual risk factors (e.g., income levels) and differences in the diagnostic capacity of the healthcare system. These findings are consistent with literature suggesting that growing income and availability of calorie-dense foods in urbanized areas intensify dietary risk exposure [47,48,49].

Overall, the identified upward trend in DM highlights the need to strengthen prevention efforts, early detection, and control of risk factors among children and adolescents, particularly in the context of rapid urbanization and changing dietary patterns characteristic of many urban areas in Kazakhstan.

The incidence of ENMD at the national level showed an overall downward trend. The most substantial decrease was observed in Almaty city, where disease indicators were among the highest. This may be explained by higher living standards and education levels, well-developed healthcare infrastructure, and better access to specialized services.

Among adolescents aged 15–17, regional patterns were more variable. For instance, a significant decline in ENMD incidence was observed in the Aktobe Region, whereas the Karaganda Region experienced an increase. This growth may be driven by several factors: first, Karaganda is known for its high level of industrial pollution, which can negatively affect children’s and adolescents’ health and contribute to endocrine and metabolic disorders. Second, socio-economic conditions—including lower income levels and limited access to quality healthcare and education compared to more developed areas such as Almaty—likely contribute to insufficient prevention and delayed diagnosis. Moreover, Karaganda may have a persistently high prevalence of risk factors, while in other regions, improved public awareness and implementation of prevention programs have helped to improve the situation [54,55,56].

Of particular interest is the observed positive correlation between the number of pediatricians and the reported incidence of ENMD in children aged 0–14. A similar direct association was found among adolescents, where an increased number of general practitioners and endocrinologists was linked to higher recorded case rates. These findings likely reflect the influence of diagnostic availability: as the number of specialists grows, so does the likelihood of identifying and diagnosing chronic endocrine and metabolic disorders in children and adolescents—a trend consistent with international research [57]. Furthermore, the rise in ENMD incidence among adolescents in certain regions may reflect a real increase in risk factors such as unhealthy diets, reduced physical activity, and increased rates of overweight, emphasizing the need for ongoing monitoring and preventive action.

Despite the substantial practical value of these findings, this study has several limitations. The use of aggregated statistical data precluded more detailed analyses by sex, ethnicity, or narrower age subgroups. Data on critical behavioral risk factors such as body mass index and obesity prevalence—key predictors of endocrine and metabolic conditions in youth—were also lacking. Furthermore, differences in data quality and completeness across regions may have influenced the interpretation of trends. Finally, due to the absence of unified, representative data on environmental determinants, potentially important environmental exposures (e.g., in industrial regions) could not be included in the analysis.

## 5. Conclusions

Overall, the observed positive trends in the reduction in iron-deficiency anemia and endocrine-metabolic disorders among children at the national level reflect improvements in the quality of preventive measures and increased access to healthcare services. At the same time, the steady rise in diabetes mellitus incidence, particularly among adolescents, indicates the ongoing impact of urbanization and changes in dietary patterns. The documented regional variability in morbidity underscores the need to design and implement territorially targeted prevention programs that consider socio-economic characteristics and the accessibility of medical services. Strengthening efforts for early identification of risk factors, promoting healthy lifestyles, and expanding access to specialized preventive care should become a priority of national health policy aimed at protecting the health of the child and adolescent population.

## Figures and Tables

**Figure 1 ijerph-22-01346-f001:**
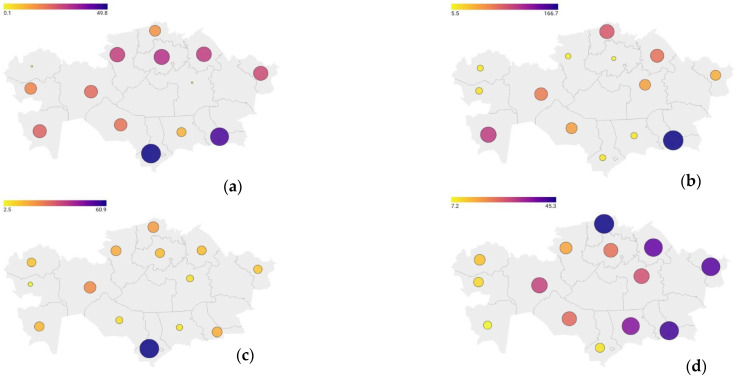
Incidence of DM among urban children population in Kazakhstan in 2013 and 2023: (**a**)—Incidence of DM among children aged 15–17 in 2013; (**b**)—Incidence of DM among children aged 15–17 in 2023; (**c**)—Incidence of DM among children aged 0–14 in 2013; (**d**)—Incidence of DM among children aged 0–14 in 2023.

**Table 1 ijerph-22-01346-t001:** Trend analysis of IDA in DM, ENMD incidence children aged 0–14 years in urban areas of Kazakhstan.

Region	IDA			DM			ENMD		
	Annual Average Change	CI (Lower; Upper)	*p*-Value	Annual Average Change	CI (Lower; Upper)	*p*-Value	Annual Average Change	CI (Lower; Upper)	*p*-Value
Kazakhstan	−285.52	−360.73; −210.31	<0.001	0.84	0.77; 1.60	<0.05	−75.68	−96.00; −55.37	<0.001
Akmola	−122.47	−209.17; −35.77	<0.05	−0.06	−1.72	0.94	−21.41	−34.01; −8.80	<0.01
Aktobe	−324.92	−418.31; −231.53	<0.001	0.33	−1.24; 1.89	0.64	−129.33	−159.26; −99.41	<0.001
Almaty	−97.72	−499.48; 304.04	0.59	1.09	−0.56; 2.74	0.17	−41.97	−115.73; 31.79	0.23
Atyrau	−22.57	−106.04; 60.89	0.56	0.79	0.39; 1.19	<0.01	−19.40	−33.19; −5.61	<0.05
West Kazakhstan	−28.75	−79.44; 21.95	0.23	0.71	−0.29; 1.71	0.14	−20.17	−32.39; −7.94	<0.01
Zhambyl	−357.01	−472.67; −241.35	<0.001	2.21	0.99; 3.42	<0.01	−49.02	−84.55; −13.49	<0.05
Karaganda	−192.48	−221.33; −163.62	<0.001	1.41	−0.50; 3.32	0.01	15.48	−19.28; 50.23	0.34
Kostanay	−105.25	−141.13; −69.36	<0.001	0.59	−0.43; 1.61	0.22	−56.82	−101.13; −12.52	<0.05
Kyzylorda	−462.8	−581.97; −343.55	<0.001	2.38	1.49; 3.27	<0.001	−65.90	−148.09; 16.29	0.10
Mangistau	−307.8	−495.98; −119.56	<0.01	−0.10	−0.93; 0.73	0.79	−15.55	−179.66; 148.56	0.83
South Kazakhstan	−591.3	−1007.96; −174.56	<0.05	−2.13	−5.41; 1.15	0.18	−165.24	−234.82; −95.66	<0.001
Pavlodar	−249.90	−317.0; −182.78	<0.01	2.25	0.97; 3.53	<0.01	−132.05	−181.57; −82.52	<0.001
North Kazakhstan	−173.59	−197.5; −149.59	<0.001	1.37	−0.64; 5.37	0.11	−85.11	−130.08; −40.14	<0.01
East Kazakhstan	−173.59	−197.5; −149.59	<0.001	1.68	−0.10; 3.47	0.06	−54.10	−86.05; −22.15	<0.01
Astana	−148.08	−208.43; −87.72	<0.001	1.96	0.22; 3.69	<0.05	−22.34	−201.81; 157.13	0.79
Almaty city	−69.69	−128.69; −10.67	<0.05	0.19	−0.14; 0.51	0.23	−274.13	−378.49; −169.78	<0.001

**Table 2 ijerph-22-01346-t002:** Trend analysis of IDA, DM, and ENMD incidence among population aged 15–17 years in urban areas of Kazakhstan.

Region	IDA			DM			ENMD		
	Annual Average Change	CI (Lower; Upper)	*p*-Value	Annual Average Change	CI (Lower; Upper)	*p*-Value	Annual Average Change	CI (Lower; Upper)	*p*-Value
Kazakhstan	−305.39	−336.96; −273.82	<0.001	1.67	0.26; 3.07	<0.05	−153.92	−192.11; −115.73	<0.001
Akmola	−200.46	−315.68; −85.23	<0.01	−1.03	−3.42; 1.36	0.35	−138.43	−218.21; −58.66	<0.01
Aktobe	−748.51	−908.32; −588.70	<0.01	2.35	−1.43; 6.13	0.19	−619.9	−751.98; −487.77	<0.001
Almaty	−214.78	−423.48; −6.09	<0.05	5.91	−2.02; 13.83	0.13	−349.6	−691.98; −7.29	<0.05
Atyrau	−118.13	−300.32; 64.06	0.18	0.71	−0.99; 2.41	0.37	10.06	−47.42; 67.53	0.70
West Kazakhstan	22.75	−10.61; 56.11	0.16	0.43	−1.16; 2.04	0.55	24.33	8.22; 40.44	<0.01
Zhambyl	−243.16	−335.94; −150.39	<0.001	1.39	−0.44; 3.23	0.12	−7.33	−149.11; 134.46	0.91
Karaganda	−98.34	−120.29; −76.39	<0.001	2.37	−1.41; 6.14	0.19	55.76	23.69; 87.83	<0.01
Kostanay	−373.85	−532.2; −215.41	<0.001	−0.28	−4.14; 3.59	0.88	−83.71	−134.88; −32.55	<0.01
Kyzylorda	−264.68	−474.59; −54.78	<0.05	5.72	1.04; 10.39	<0.05	−287.7	−558.93; −16.55	<0.05
Mangistau	−801.52	−972.23; −630.79	<0.001	3.31	−2.17; 8.79	0.21	−80.82	−211.43; 49.79	0.19
South Kazakhstan	−548.20	−964.99; −131.42	<0.05	−1.86	−4.71; 0.99	0.17	−123.1	−207.96; −38.32	<0.01
Pavlodar	−148.93	−224.88; −72.99	<0.01	2.75	−1.42; 6.92	0.17	−420.2	−527.48; −313.02	<0.001
North Kazakhstan	−102.98	−142.16; −63.80	<0.001	9.64	2.41; 16.86	0.012	−57.19	−142.23; 27.85	0.16
East Kazakhstan	−249.73	−304.28; −195.18	<0.001	−0.63	−4.47; 3.20	0.71	−158.47	−311.10; −5.85	<0.05
Astana	−258.1	−547.96; 3.18	0.07	2.29	−0.73; 5.33	0.12	−88.44	−289.54; 112.68	0.35
Almaty city	−381.56	−539.02; −224.09	<0.001	−0.68	−1.96; 0.60	0.26	−297.35	−502.05; −92.66	<0.01

**Table 3 ijerph-22-01346-t003:** Descriptive statistics of dependent variables.

Variables	Mean	Minimum	Maximum	Std. Deviation
incidence of IDA in children 0–14 years of age in urban areas (per 100,000 population)	2396.8744	253.00	7421.10	1704.34813
incidence of IDA in adolescents 15–17 years old in urban areas (per 100,000 population)	2526.1010	111.40	9957.60	1972.51857
incidence of endocrine system diseases, nutritional and metabolic disorders in children 0–14 years of age in urban areas (per 100,000 population)	1047.8030	112.70	3911.20	738.54446
incidence of ENMD in adolescents 15–17 years of age in urban areas (per 100,000 population)	2196.8884	163.30	8234.90	1447.70057
incidence of DM in children 0–14 years of age in urban areas (per 100,000 population)	16.5477	2.50	60.90	9.46182
incidence of DM in adolescents 15–17 years old in urban areas (per 100,000 population)	23.6471	0.00	166.70	22.54657

**Table 4 ijerph-22-01346-t004:** Descriptive statistics of independent variables.

Variables	Mean	Minimum	Maximum	Std. Deviation
The average share of the urban population with incomes below the subsistence minimum, %	2.46	1.20	4.10	1.25
Gini coefficient quantifies income inequality on a scale from 0 (perfect equality) to 1 (maximum inequality)	0.24	0.16	0.32	0.04
The average household monetary expenditures per capita, in thousands KZT	55.20	22.63	12.14	20.70
Household income used for consumption by region, thousand KZT	57.16	24.13	12.14	20.63
The average per capita food expenditures, in thousands KZT	13.18	3.44	33.38	6.41
The number of specialized therapists	16.20	6.10	30.60	5.13
The average number of general practitioners per 10,000 urban population	5.80	1.80	10.90	1.64
The average number of endocrinologists	0.60	0.20	1.40	0.22
The average number of pediatricians	3.84	2.10	8.10	1.06
Number of organizations with children’s outpatient departments and offices per 1000 population	107.29	7.00	1337.00	220.08
The availability of pediatricians (including neonatologists) for the 0–14 age group per 1000 children	0.82	0.19	2.65	0.41
The average availability of pediatric hospital beds of all profiles for the 0–14 age group per 1000 children	3.99	1.91	8.01	1.16

**Table 5 ijerph-22-01346-t005:** Multiple linear regression predicting incidences of iron-deficiency anemia, DM, and ENMD among children 0–14 years and adolescents 15–17 years.

Predictor	B	Std. Error	Sig.	95.0% Confidence Interval for B		R^2^
				Lower Bound	Upper Bound	
Incidence of IDA in children 0–14 years of age in urban areas (per 100,000 population)						
The average per capita food expenditures, in thousands KZT	−203.452	22.250	0.000	−253.785	−153.120	0.903
Incidence of IDA in adolescents 15–17 years of age in urban areas (per 100,000 population)						
The average per capita food expenditures, in thousands KZT	−215.967	10.450	0.000	−239.608	−192.327	0.979
Incidence of endocrine system diseases, nutritional and metabolic disorders in children 0–14 years of age in urban areas (per 100,000 population)						
The average number of pediatricians	890.987	70.120	0.000	732.364	1049.609	0.947
Incidence of endocrine system diseases, nutritional and metabolic disorders in adolescents 15–17 years of age in urban areas (per 100,000 population)						
The number of specialized therapists	603.730	64.459	0.000	455.088	752.372	0.950
The average number of general practitioners per 10,000 urban population	801.574	319.884	0.037	63.920	1539.228	
Incidence of DM in children 0–14 years of age in urban areas (per 100,000 population)						
The average household monetary expenditures per capita, in thousands KZT	0.489	0.092	0.001	0.000	0.001	0.844
The average number of general practitioners per 10,000 urban population	5.259	1.504	0.008	1.791	8.727	
Incidence of DM in adolescents 15–17 years of age in urban areas (per 100,000 population)						
Household income used for consumption by region, thousand KZT	0.379	0.038	0.000	0.289	0.469	0.960
The average availability of pediatric hospital beds of all profiles for the 0–14 age group per 1000 children	19.314	2.677	0.000	12.985	25.644	
The average number of endocrinologists	56.508	14.876	0.007	21.332	91.683	
Tolerance > 10; VIF < 4						

## Data Availability

The data presented in this study are available on request from the corresponding author.

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
