# Peer review of "Diabetes, Iron-Deficiency Anemia, and Endocrine, Nutritional, and Metabolic Disorders in Children: A Socio-Epidemiological Study in Urban Kazakhstan"

_ijerph, 2025, doi:10.3390/ijerph22091346_

Round 1
Reviewer 1 Report
Comments and Suggestions for Authors
Overall comments - interesting topic but methods and results are incomplete or not adequately justified. For example, How DM, IDA, and ENMD were defined is not defined anywhere in the paper. How the multiple regression models were specified was also not described.
Any analysis of IDA would need to account for sex.
To make the findings meaningful, additional context is required (for example, a "1,000 tenge increase in HH food expenditure" in the abstract was correlated with improved health, but that increased spending on food covers what time frame?)
Further, generalized statements, such as "Regional disparities were evident and closely linked to socio-economic factors." are not actionable and not very meaningful. Suggest revising to be more pragmatic about specific findings.
This concluding statement, "... the importance of targeted, region-specific prevention programs, early risk identification, and addressing structural inequalities in child and adolescent healthcare amidst ongoing urban and nutritional transitions." is not novel, although some of the results were and may warrant more attention in the discussion section.
Author Response
Overall comments - interesting topic but methods and results are incomplete or not adequately justified.
Comment 1: For example, How DM, IDA, and ENMD were defined is not defined anywhere in the paper.
Response 1: Thank you for your comment. In the Republic of Kazakhstan, data collection on the primary incidence of type 1 and type 2 diabetes mellitus, iron deficiency anemia, and other metabolic disorders among children is regulated by official directives of the Ministry of Health, particularly Order No. 128 dated March 6, 2013 (as amended), which approves the standardized forms of ad-ministrative health data. Healthcare institutions are required to record newly diagnosed cases in accordance with ICD-10 classifications (including E10–E14 for diabetes and D50 for iron deficiency anemia), specifying the patient’s age, region, and, where available, sex. Data collection is conducted by attending physicians during outpatient or inpatient visits and is entered into electronic medical information systems. Aggregated data are subsequently submitted to regional public health authorities and then to the Ministry of Health. Each year, by February 10, all healthcare providers must submit their summary reports using Form No. 128 as pre-scribed.
The calculation of primary morbidity indicators is carried out based on the methodology approved by the Ministry of Health, whereby the incidence is expressed as the number of newly registered cases per 100,000 children in the population. Finalized data are published through official platforms such as the Bureau of National Statistics and its specialized child health portal, as well as in the annual statistical compendium "Healthcare of Kazakhstan and Activities of Healthcare Organizations." However, disaggregation of data by sex and age is sometimes limited, as some publicly available datasets are presented in aggregated form.
Comment 2: How the multiple regression models were specified was also not described.
Response 2: Thank you for your valuable feedback. In response to your comment regarding the specification of the multiple regression models, we have expanded the Methods section to clarify this aspect. Specifically, we now include a description of how the multiple linear regression models were specified. Each model was constructed with a single dependent variable (the incidence of IDA, DM, or ENMD in either age group) and a set of predefined independent variables related to socioeconomic status and healthcare availability. Model selection was based on theoretical relevance and empirical correlation, and all predictor variables were entered simultaneously (enter method). Multicollinearity was assessed using tolerance and variance inflation factor (VIF) statistics, ensuring that all included variables met acceptable thresholds (tolerance > 0.1; VIF < 4). Separate models were developed for each outcome and age group, resulting in six models in total. These clarifications have been integrated into the revised manuscript.
Comment 3: Any analysis of IDA would need to account for sex.
Response 3: We acknowledge that the analysis of sex-specific trends in iron deficiency anemia (IDA) among children was not included in this study due to the unavailability of disaggregated data in official statistical sources. This limitation is recognized as one of the methodological constraints of our research.
Comment 4: To make the findings meaningful, additional context is required (for example, a "1,000 tenge increase in HH food expenditure" in the abstract was correlated with improved health, but that increased spending on food covers what time frame?)
Response 4: Thank you for your note. We added time frame information into abstract
Comment 5: Further, generalized statements, such as "Regional disparities were evident and closely linked to socio-economic factors." are not actionable and not very meaningful. Suggest revising to be more pragmatic about specific findings.
Response 5: We appreciate the reviewer’s constructive feedback. In response, we have revised the abstract to enhance the clarity, specificity, and analytical depth of our findings. We have aimed to ensure that the interpretations are more pragmatic and grounded in the data
Comment 6: This concluding statement, "... the importance of targeted, region-specific prevention programs, early risk identification, and addressing structural inequalities in child and adolescent healthcare amidst ongoing urban and nutritional transitions." is not novel, although some of the results were and may warrant more attention in the discussion section.
Response 6: We appreciate the reviewer’s constructive feedback. In response, we have revised the abstract to enhance the clarity, specificity, and analytical depth of our findings. We have aimed to ensure that the interpretations are more pragmatic and grounded in the data
Reviewer 2 Report
Comments and Suggestions for Authors
Thank you for the interesting report - I have read it in one breath. It is important to keep an eye on the trends in DM, IDA and metabolic disease in children. As authors note themself, the results are not surprising and largely follow the trends observed in different countries. Hope you have a chance to present your work to policy makers in your country, as an evidence towards the need to invest in monitoring, education on and prevention of DM in particular, and in support of further work on IDA prevention programmes.
I was wondering if you have a comment on decrease in metabolic disease in relation to decrease of IDA - taking into account iron - metabolic disease link?
I was hoping that there would be exploration of link with industrial pollution, and indeed I note that you mention in conclusion this is a possibility but data is not available - would be important to explore this once/where the data is available (follow on paper perhaps).
Few suggestions related to readability of the paper:
- it would be good to abbreviate some of the variables. for example: "Number of pediatricians per 10,000 urban population" could become "pediatricians no." or "Provision of children's beds of all profiles per 1,000 children aged 0–14 years" becomes "hospital beds" - this would help readability of results and discussions.
- Figure 1 would make more sense is left side is 2013 (a and c) and right side is 2023 (b and d)
- Table 4 you have a line with cyrillic writing
Wishing you luck with submission!
Author Response
Thank you for the interesting report - I have read it in one breath. It is important to keep an eye on the trends in DM, IDA and metabolic disease in children. As authors note themself, the results are not surprising and largely follow the trends observed in different countries. Hope you have a chance to present your work to policy makers in your country, as an evidence towards the need to invest in monitoring, education on and prevention of DM in particular, and in support of further work on IDA prevention programmes.
Response: Thank you for you assessment and support!
Comment 1: I was wondering if you have a comment on decrease in metabolic disease in relation to decrease of IDA - taking into account iron - metabolic disease link?
Response 2: We appreciate the reviewer’s thoughtful question regarding the observed concurrent declines in IDA and ENMD. A substantial body of evidence supports a physiological link between iron status and endocrine/metabolic function. For instance, iron deficiency impairs thyroid hormone synthesis by reducing thyroid‑peroxidase activity and diminishing conversion of Tâ‚„ to T₃, which can lead to subclinical or overt hypothyroidism. Similarly, Mendelian randomization studies indicate a causal association between hypothyroidism and increased risk of IDA (OR ~1.10, p < 0.001). Moreover, in pediatric populations with obesity and insulin resistance, disrupted iron homeostasis and chronic inflammation are tied to both IDA and metabolic dysregulation, suggesting a bidirectional relationship between iron deficiency and metabolic health outcomes. These findings mirror our national observation that reductions in IDA were paralleled by declines in broader endocrine, nutritional, and metabolic disorders. The epidemiological and clinical evidence supports the hypothesis that improvements in iron status may contribute to systemic metabolic stabilization, and vice versa. We have added a brief discussion of these mechanistic and clinical insights in the revised manuscript to contextualize the link between IDA and ENMD trends.
Comment 2: I was hoping that there would be exploration of link with industrial pollution, and indeed I note that you mention in conclusion this is a possibility but data is not available - would be important to explore this once/where the data is available (follow on paper perhaps).
Response 2: We greatly appreciate the reviewer’s insightful observation regarding the potential role of industrial pollution in the incidence of childhood endocrine and metabolic disorders. Indeed, environmental exposures—including air and soil pollution from industrial sources—are increasingly recognized as important determinants of pediatric health outcomes, particularly in urban settings. However, as noted in the manuscript, the primary aim of our study was to assess the association between socio-economic and health system factors and the incidence of iron-deficiency anemia, diabetes mellitus, and ENMD among children and adolescents in urban areas of Kazakhstan. At present, regionally disaggregated and age-specific data on environmental exposures, including industrial pollution, are limited or not publicly available in Kazakhstan. This constrained our ability to integrate such variables into the current analytical framework. Nevertheless, we fully agree with the reviewer that this is a highly relevant and important area for future research. Once reliable environmental exposure data become available, follow-up studies examining the interaction between socio-economic vulnerability and environmental risk factors—including industrial pollution—will be essential to fully understand the multifactorial determinants of pediatric disease trends in urban Kazakhstan.
Comment 3: it would be good to abbreviate some of the variables. for example: "Number of pediatricians per 10,000 urban population" could become "pediatricians no." or "Provision of children's beds of all profiles per 1,000 children aged 0–14 years" becomes "hospital beds" - this would help readability of results and discussions.
Response 3: We agree that simplifying and abbreviating some of the variable names can enhance clarity and make the results and discussion sections more accessible to readers. Accordingly, we have revised the variable labels in the text, tables, and figures.
Comment 4: Figure 1 would make more sense is left side is 2013 (a and c) and right side is 2023 (b and d)
Response 4: We agree that presenting the panels with 2013 data on the left (a and c) and 2023 data on the right (b and d) improves the logical flow and visual interpretation of the figure. We have adjusted the panel order accordingly and updated the figure legend to reflect this change.
Comment 5: Table 4 you have a line with cyrillic writing
Response 5: We have reviewed Table 4 and identified the line containing Cyrillic text. This was included unintentionally during the drafting process. We have now corrected the table to ensure consistency in language throughout the manuscript.
Comment 6: Wishing you luck with submission!
Response 6:
Thank you very much for your encouraging words and thoughtful feedback throughout. We sincerely appreciate your time and valuable suggestions, which have helped us improve the clarity and quality of the manuscript.
Round 2
Reviewer 1 Report
Comments and Suggestions for Authors
Thank you for providing thoughtful revisions to the manuscript. The additional details on the source of the administrative data and the consistency in defining conditions at the national level is very useful. Two minor points remain that could make the paper more useful. One is to indicate that the linear regression are ecological models, so that it is very clear that the analyses are not at the individual level but at a regional level (to specify). The second point would be that the modifications made in the abstract (in response to points 5 & 6 of the original review) should also be mirrored in the discussion. For example, lines 365-366: in the discussion, not only the abstract, indicate a time period for " an increase of 1,000 tenge in per capita food expenditures is associated with a decrease of 203–216 cases of IDA per 100,000 children" (annually?). Finally, line 194 of the revision that describes the predictor variables entered simultaneously ("enter method") seems incomplete and requires attention.
Author Response
Comment 1: Thank you for providing thoughtful revisions to the manuscript. The additional details on the source of the administrative data and the consistency in defining conditions at the national level is very useful. Two minor points remain that could make the paper more useful.
One is to indicate that the linear regression are ecological models, so that it is very clear that the analyses are not at the individual level but at a regional level (to specify).
Response 1: We thank the reviewer for this important remark. We have clarified in the Methods section that the linear regression analyses were conducted as ecological models at the regional level, not the individual level. The corresponding revision has been made in the manuscript.
Comment 2: The second point would be that the modifications made in the abstract (in response to points 5 & 6 of the original review) should also be mirrored in the discussion. For example, lines 365-366: in the discussion, not only the abstract, indicate a time period for " an increase of 1,000 tenge in per capita food expenditures is associated with a decrease of 203–216 cases of IDA per 100,000 children" (annually?).
Response 2: We thank the reviewer for this valuable comment. We have revised the Discussion accordingly and now specify the time period (2013–2023) in which the association between per capita food expenditures and IDA cases was observed.
Comment 3: Finally, line 194 of the revision that describes the predictor variables entered simultaneously ("enter method") seems incomplete and requires attention.
Response 3: Thank you for your comment. We made corrections.